# Development and Survival of Human Ovarian Cells in Chitosan Hydrogel Micro-Bioreactor

**DOI:** 10.3390/medicina58111565

**Published:** 2022-10-31

**Authors:** Elsa Labrune, Cyrielle Fournier, Benjamin Riche, Laurent David, Alexandra Montembault, Sophie Collardeau-Frachon, Mehdi Benchaib, Jacqueline Lornage, Jean Iwaz, Bruno Salle

**Affiliations:** 1Université de Lyon, F-69000 Lyon, France; 2Université Claude Bernard Lyon 1, F-69100 Villeurbanne, France; 3Service de Médecine de la Reproduction et Préservation de la Fertilité—CECOS, Hôpital Femme Mère Enfant, Hospices Civils de Lyon, F-69500 Bron, France; 4Institut Cellule Souche et Cerveau (SBRI), INSERM U 1208, F-69500 Bron, France; 5Service de Biostatistique-Bioinformatique, Pôle Santé Publique, Hospices Civils de Lyon, F-69003 Lyon, France; 6Équipe Biostatistique-Santé, Laboratoire de Biométrie et Biologie Évolutive, CNRS UMR 5558, F-69100 Villeurbanne, France; 7L’Institut National des Sciences Appliquées de Lyon, F-69100 Villeurbanne, France; 8Université Jean Monnet, F-42100 Saint-Etienne, France; 9Ingénierie des Matériaux Polymères (IMP), CNRS UMR 5223, F-69100 Villeurbanne, France; 10Service D’anatomopathologie, Groupement Hospitalier Est, Hospices Civils de Lyon, F-69500 Bron, France; 11Faculté de Médecine Lyon Sud, F-69600 Oullins, France

**Keywords:** chitosan, hydrogel, folliculogenesis, bioreactor, tissue culture technique

## Abstract

*Background and Objectives:* To test the long-term ability of human ovarian cortex cells to develop in unconventional culture conditions. *Materials and Methods.* Ovarian cortex cells from fetuses aged 23 to 39 weeks gestation were cultured for 90 days in hollow chitosan hydrogel micro-bioreactors and concurrently in traditional wells. Various cell-type counts were considered. *Results*: With intact follicles as a denominator, the percentage of growing intact follicles at Day 0 varied widely between ovaries (0 to 31.7%). This percentage tended to increase or stay relatively constant in bioreactor as in control cultures; it tended more toward an increase over time in bioreactor vs. control cultures. Modeled percentages showed differences (though not significant) in favor of bioreactor cultures (16.12% difference at D50 but only 0.12% difference at D90). With all follicles present as a denominator, the percentage of growing primary and secondary follicles at D0 varied widely between ovaries (0 to 29.3%). This percentage tended to increase over time in bioreactor cultures but to decrease in control cultures. Modeled percentages showed significant differences in favor of bioreactor cultures (8.9% difference at D50 and 11.1% difference at D90). At D50 and D90, there were only few and sparse apoptotic cells in bioreactor cultures vs. no apoptotic cells in control cultures. *Conclusions*: Over three months, bioreactor folliculogenesis outperformed slightly traditional culture. This is an interesting perspective for follicle preservation and long-term toxicological studies.

## 1. Introduction

Today, cryopreservation of the ovarian cortex is the only method able to preserve the fertility of prepubescent girls and women in case of an urgent need for chemotherapy [1,2]. The ovary is surgically removed, then its cortex frozen until the use of its cells for in vivo transplantation. Since 2004, this innovative technique allowed giving birth to nearly 200 children worldwide [3,4].

The major disadvantage of using cryopreserved ovarian tissue for fertilization is the risk of transfer of malignant cells present in the autograft [5]. Thus, it is essential to develop a new strategy for fertility preservation through obtaining sufficient mature oocytes to enable successful implantation and pregnancy. One promising technique is to obtain mature oocytes through in vitro folliculogenesis, a method that succeeded and led to live births in mice [6,7]. In humans, only two teams were able to obtain mature oocytes from unilaminar follicles [8,9]. McLaughlin et al. obtained mature oocytes after 21 days: 8 days for the culture of ovarian tissue fragments and 13 days for the culture of isolated follicles from those fragments (vs. ~300 days in the ovary) but these oocytes were atypical, and the culture yield was very low [8]. Xu et al. reported very recently having obtained mature oocytes after 63 days: 21 days for ovarian tissue fragments and 42 days for follicles isolated from those fragments. The first phases of ovarian tissue culture (i.e., 8 and 21 days) were traditional, i.e., without any specific unconventional culture system [9] and might have been stopped because of tissue alteration over time. Those results were very promising but oocyte developmental competence via in vitro fertilization could not be demonstrated. The latter process was complex because of the need for multiple steps and for follicle isolation.

Today, there are several three-dimensional (3D) culture systems that imitate the ovarian gland and provide mature gametes of better quality in humans [10,11,12,13] or animals [14,15], but no mature follicles from those ‘artificial ovaries’ led to mature oocytes and no births were reported in farm animals [16]. The use of 3D culture systems seems necessary to maintain cellular interactions, obtain 3D follicle conformation, and ensure long growth periods [17] (about three months). 

We designed a 3D culture system consisting in a chitosan tubular bioreactor that proved able to host a fragment of the ovarian cortex and promote its growth. We report on the development and survival of that tissue over three months of culture. 

The success of such a culture technique will facilitate follicle culture (in a single step), help obtain follicles of various stages for (i) pharmacological or toxicological tests; (ii) fertility preservation before radiation or chemotherapy treatment, or, (iii) medically-assisted procreation. The technique might be also adapted to help assisted animal reproduction.

## 2. Materials and Methods

### 2.1. Chitosan Characterisation

Chitosans constitute a family of diverse polymers (all linear copolysaccharides of β(1-4)-linked N-acetyl-glucosamine and N-glucosamine) that result from deacetylation of chitin. For the sake of reproducibility, we provide a detailed characterization of the chitosan(s) used, the mean degree of deacetylation (i.e., the mean fraction of N-acetyl-D-glucosamine residues in the sample), and the molar mass distribution parameters such as the weight-average molar mass (Mw) and the number-average molar mass (Mn). 

For the bioreactors, we used a high-grade chitosan obtained from squid pens chitin (Mahtani Chitosan Pvt. Ltd., Veraval, India; batch type 114). The SEC-MALLS analysis of this batch indicated a weight-average molecular weight (Mw) of 630 ± 50 kg/mol and a polydispersity index (Mw/Mn) of 1.6. The mean degree of acetylation, as determined by ^1^H spectroscopy [18], was 5 ± 1%.

### 2.2. Chitosan Bioreactor Preparation

Chitosan hydrogel bioreactors were prepared as previously described by Rivas Araiza et al. in 2007 [19] and 2008 [20].

Chitosan was first purified according to Montembault et al. [21] (2005) then chitosan hydrogels were prepared by gelation of an aqueous chitosan acetate solution. Briefly, purified chitosan was dissolved in an acetic acid solution at a polymer concentration of 0.5% and then filtered successively through 5, 3, 0.8, and 0.42-µm filters (Millipore, Molsheim, France). Filtered chitosan acetate was next precipitated with ammonia 20–30% *w/w* (Sigma-Aldrich, Saint-Quentin-Fallavier, France), washed with deionized water, and freeze-dried.

For bioreactor production, a chitosan solution at 2% *w/w* polymer concentration was obtained by dispersion of freeze-dried chitosan in water. Acetic acid (Sigma-Aldrich) was added in stoichiometric amounts to achieve the protonation of amine groups of glucosamine residues. That solution was centrifuged in a dispenser syringe (Nordson EFD, Serris, France), then extruded through a 6-mm diameter tip using a Performus I dispenser (Nordson EFD). The extrudate was neutralized for 2 min in sodium hydroxide 1M. This neutralisation coagulated the surface of the extrudate and resulted in a tubular hydrogel that was stabilized in deionized water. The uncoagulated inner chitosan acetate solution was removed by air insufflation creating thus a single hollow tube from which 3 or 4 portions were cut to constitute independent bioreactors. These bioreactors had a 4-mm inner diameter and 1-mm wall thickness (Figure 1). They were sterilized in water flasks by autoclave (liquid protocol) for 20 min at 121 °C and stored until use in deionized water at room temperature.

### 2.3. Ovarian Tissue Culture

Fresh ovaries (*n* = 5) extracted from human fetuses aged 23 to 39 weeks gestation were transferred from the tissue biobank to the laboratory in a culture medium (MEM Alpha Medium^®^, Corning Life Sciences, Tewksbury, MA, USA) at 20–22 °C within 30 min of collection. Each ovary was split into five equivalent volume/weight fragments (~2 × 2 × 2 mm and 20 ± 7 mg) of which one was fixed in 4% formaldehyde (VWR, Strasbourg, France) for 24 h before being embedded in paraffin (VWR). From the remaining four fragments, (i) two were introduced each in a bioreactor that was then tied at each end with suture thread and immersed in a well-containing culture medium (= ‘bioreactor’ culture) (Figure 1); and, (ii) two were immersed free in similar wells and similar culture medium (=control cultures). 

The Alpha Medium of all cultures (minimum essential medium—MEM; Corning Life Sciences, NY, USA) contained L-glutamine (Corning Life Sciences), 10% serum substitute supplement (SSS, Irvine Scientific, Santa Ana, CA, USA), 100 IU/mL penicillin G (HyClone, South Logan, UT, USA), and 100 µg/mL streptomycin (HyClone), 2.5 µg/mL amphotericin B (PAN-Biotech, Bayern, Germany), 10 ng/mL insulin (PAN-Biotech), 10 ng/mL transferrin (PAN-Biotech), 1 ng/mL selenium (PAN-Biotech), 25 mIU/mL rFSH (Bemfola^®^, Gedeon Richter, Budapest, Hungary), and 50 µg/mL ascorbic acid (Sigma-Aldrich). 

Bioreactor-and control cultures had to be maintained for 90 days at 37 °C in humidified air with 5% CO_2_. Every other day, half of the culture medium of each well was removed and replaced by fresh medium. At 50 and 90 days of culture, one specimen from each type of culture was fixed in 4% formaldehyde (VWR) for 24 h before being embedded in paraffin (VWR).

### 2.4. Histological Analyses

Paraffin-embedded fragments were cut into 4-µm serial slices. From each series, every third slice (up to 25 slices) was stained with hematoxylin (Millipore), eosin (Sigma-Aldrich), and safran (RAL diagnostics, Martillac, France) and examined under MOTIC microscope Panthera (Panthera CC^®^, Motic Europe, Barcelona, Spain) for follicular density and morphology allowing classification.

Follicular density was defined as the number of follicles per mm^2^. The follicles were classified according to Gougeon [22] (1956). That classification considers five successive development stages: (i) Primordial stage: oocyte surrounded by flattened granulosa cells; (ii) Intermediary stage: oocyte surrounded by flattened and at least one cuboidal granulosa cell; (iii) Primary stage: oocyte surrounded only by cuboidal granulosa cells; (iv) Secondary stage: oocyte surrounded by ≥2 layers of cuboidal granulosa cells, and, (v) Tertiary stage: follicle with single antral cavity [22].

Regarding quality, altered follicles were those with at least one of the following signs of oocyte or granulosa cell degeneration: (i) presence of pyknotic oocyte or follicular cell nuclei; (ii) detachment of the oocyte from the surrounding granulosa cells; (iii) vacuolization of the ooplasm; (iv) partial degeneration of granulosa cells, or, (v) detachment of the basal membrane. Otherwise, follicles were considered ‘intact’ [23].

For the assessments of follicular density, follicular classification [22], growth, and survival [23], each slide was examined by two independent readers, and the results were averaged.

### 2.5. Immunostaining of Ovarian Tissue Slices

Ovarian tissue slices destined for immunostaining were first dewaxed in methylcyclohexane (Sigma-Aldrich) and absolute ethanol (Sigma-Aldrich) and then rinsed with water. Unmasking of antigen epitopes was done by incubation of slide-mounted slices for 45 min in a sodium citrate solution pH = 6 (Merck, Darmstadt, Germany) preheated and maintained at a sub-boiling temperature in a water bath.

Proliferation- or apoptosis-specific antigen detection used kits with horseradish peroxidase and 3,3′-diaminobenzidine (DAB) as chromogen (EnVisionTM G/2 Doublestain System^®^, Dako, Glostrup, Denmark). Before antibody use, cell endogenous enzymes were blocked with a kit-included preparation.

Tissue slices destined for proliferation assays were incubated for one hour with a rabbit monoclonal anti-Ki67 antibody, a specific marker of proliferating cells (#9027; Cell Signaling Technology, Danvers, MA, USA). 

Tissue slices destined for apoptosis assays were incubated for one hour with a rabbit monoclonal anti-cleaved caspase-3 antibody, a specific marker of apoptotic cells (#9664S; Cell Signaling Technology).

After those incubations, immunostaining was visualized by incubation for five minutes with DAB+ Chromogen (3,3′-diaminobenzidine in chromogen solution) (EnVision^TM^ G/2 Doublestain System^®^, Dako, Glostrup, Denmark). Counterstaining used standard hematoxylin (Millipore). 

Control slices for cell proliferation and apoptosis were obtained from fresh ovarian tissue and thymic tissue, respectively (the latter being known for its high percentage of apoptotic cells).

For microscopic observation, the slices were mounted with an aqueous mounting medium (Dako Faramount^®^, Dako). The independent readers carried out the microscopic observation to assess densities of proliferating and apoptotic cells by counting, respectively, the number of DAB+ and brown cells per field, averaging the counts over five fields, and expressing the results in the number of cells/mm².

### 2.6. Statistical Analyses

The analysis proceeded on the assumption of ‘independent and identically distributed’ observations. More precisely, the basic assumptions were that: (i) the specimens submitted to culture would be homogeneous (regular cell structure within each specimen) and exchangeable (identical structures between specimens), and (ii) there would be a difference between the processes of follicle growth according to the type of culture (bioreactor vs. control). 

In the first set of analyses, the percentages of follicles undergoing growth (i.e., primary and secondary follicles) were calculated according to the culture time and type considering the number of intact follicles as the denominator. In these conditions, the percentage of intact follicles undergoing growth was defined as (number of intact primary follicles + number of intact secondary follicles)/total number of intact follicles × 100.

In the second set of analyses, the percentages of follicles undergoing growth (i.e., primary and secondary follicles) were calculated according to the culture time and type considering all follicles present as the denominator. In these conditions, the percentage of intact follicles undergoing growth was defined as (number of intact primary follicles + number of intact secondary follicles)/total number of follicles × 100.

In a modelling approach, all pieces of information, including on D0 were used to compare the results of the two culture types. The chosen model was a linear regression model with a random effect put on the intercept. This model included four indicator variables (as covariates) to represent the four culture conditions and a single random term (intercept) per ovary to take into account the very high level of heterogeneity of the proportions at D0. One model was run per set of proportions as calculated in each set of analyses. All estimations were made using the maximum likelihood method.

### 2.7. Legal and Ethical Considerations

All study procedures met the relevant ethical guidelines and obtained the approval of the local ethics committee. Written consent to store and use the tissues for research was obtained from the parents. The samples were provided by the local tissue bank (Centre de Ressources Biologiques des Hospices Civils de Lyon; n° BB-0033-00046). The experimental protocol was approved (approval AC 2019-3465, June 2019) by the Centre de Ressources Biologiques—Hospices Civils de Lyon (CRB-HCL), an organization registered with the Ministère de l’Enseignement Supérieur et de la Recherche (ministerial authorisation DC-2011-1437 CRB HCL).

## 3. Results

At D0, nearly 72% of fresh tissue follicles were primordial follicles; there were also a few intermediate and primary follicles but no secondary follicles (Figure 2). It is important to note that no tertiary follicles were seen at any time in any culture.

### 3.1. Results of the First Set of Analyses with Intact Follicles as Denominator

At D0, the percentage of growing follicles (pGF) varied widely between ovaries (0 to 31.7%). Table 1 shows that the pGF tended to increase or remain practically stable over time, either in bioreactor or control cultures (D50 and D90 vs. D0) (Figure 3 and Figure 4). 

The results indicated that the percentages of growing follicles tended to increase in bioreactor more than in control cultures and that three ovaries displayed better dynamics than the remaining two.

Percentage comparisons showed a 16.12% difference at D50 but only a 0.12% difference at D90. These differences were in favor of bioreactor cultures, but they were not statistically significant because of wide 95% confidence intervals (−34.12; 1.9 and −16.2; 16.4 for bioreactor and control cultures, respectively).

### 3.2. Results of the Second Set of Analyses with All Present Follicles as Denominator

In the second set of analyses, using normalization by the number of all present follicles, the percentage of growing follicles varied widely between ovaries at D0 (0 to 29.3%). Table 2 shows that the fraction of growing follicles tended to increase over time in bioreactor cultures (D50 and D90 vs. D0) but to decrease in control cultures. Fetal ovarian tissue histology slices that illustrate these trends are given in Table 3 and Figure 5 and Figure 6). 

Comparisons between modeled percentages showed an 8.9% difference at D50 and an 11.1% difference at D90, both in favor of bioreactor cultures. The 95% confidence intervals of those two differences (−16.5; −1.27 and −18.7; −3.5, respectively) excluded value 0 and allow rejecting the null hypothesis; i.e., the percentages of growing follicles at D50 and D90 were significantly higher in bioreactor than in control cultures.

### 3.3. Immunostaining for Cell Apoptosis

At D50 and D90, there were only few and sparse apoptotic cells in bioreactor cultures (6.8 ± 2.2 and 2.2 ± 1.1 cleaved caspase-3-positive cells/mm^2^, respectively), whereas there were no apoptotic cells in control cultures (Table 3 and Figure 7).

## 4. Discussion

In this work, we assessed the growth of human ovarian cortical tissue in a chitosan bioreactor (a closed hydrogel tube). The porous structure of the hydrogel (98% *w/w* water) enabled the diffusion of the components of the culture medium to the cells but kept paracrine, juxtacrine, and autocrine secretions in the bioreactor after culture medium renewals. We observed a spontaneous initiation of folliculogenesis and a maturation from the primordial to the secondary follicle stage. Overall, that initiation was quantitatively and qualitatively more efficient in bioreactors than in classical control cultures. Furthermore, in the bioreactor, cell proliferation continued steadily over three months. To our knowledge, the most currently available literature has never reported on such a long period of ovarian tissue culture.

This study shows a slow and spontaneous initiation of folliculogenesis in the bioreactor. This activation was slower than what was already described [8] and rather close to the natural physiological pace. Indeed, the specialized literature has reported that, in vitro, human primordial follicles transform into secondary follicles over 14.8 ± 11.2 days [8,9,10,11,12,24,25,26,27,28,29,30,31,32,33,34,35], which is a very short period in comparison with the physiological transformation that lasts nearly 300 days [22]. Furthermore, in previous studies, growing follicles were extracted and then cultured, in biomaterial beads or without matrix, before oocyte isolation for an in vitro maturation [36]. The latter step was needed because no system was available for the long survivals of large amounts of tissue. The initiation of oocyte activation could have been tissue fragmentation by itself [37]. However, this might not be a serious problem because the aim of the experiments was the length and issue of cultured oocytes rather than the cause or promotion of their activation.

The bioreactor culture technique allowed comparing cell proliferation and apoptosis between bioreactor and control cultures over three months. It ensured a higher density of proliferating cells (vs. fresh tissue) and a higher density of apoptotic cells. One explanation for that higher density of apoptotic cells is that, in control cultures, follicles are altered but not eliminated, whereas, in bioreactor cultures, altered follicles are eliminated and, therefore, are no longer visible on histological examination. Another explanation may be the wide heterogeneity of the ovarian tissues cultured: those put in the bioreactors would be less rich in follicles than those put in a traditional medium. Checking each of those explanations requires non-negligible amounts of time and resources but is needed for definitive conclusions about the potential benefits of bioreactors. Anyway, the present report has shown that ovarian fragments of the same size and weight may have already heterogeneous cell structures and include very different follicle densities.

Culture systems for isolated follicles are now proposed to test the effects of toxic chemicals [38]. Unfortunately, studies carried out on isolated follicles would provide only partial answers because ovarian functioning and folliculogenesis do not depend only on the follicles but also on the surrounding tissues (stroma, other follicles, and the oocyte). This justifies maintaining all cell interactions needed for folliculogenesis by cultivating samples of entire ovarian tissue [17]. Furthermore, numerous tests require live ovarian tissue for weeks. Within this context, bioreactor cultures may represent a paradigm change because the three-month persistence of sufficient and viable cells offers sufficient time to test potential endocrine disrupters on ovarian tissue or the pharmacological properties of candidate drugs for chemotherapy or targeted therapy against cancer.

This preliminary study has shown that, under the herein-described tentative conditions, bioreactor cultures were more performant than traditional cultures. Proportion changes were most probably due to follicle growth because we did not observe much excess cell mortality between the two delays. One major explanation for that higher performance is that the ‘chamber’ created around the specimen put in culture would retain the paracrine factors that enhance cell growth, activation, or differentiation.

One interesting fact was the presence of apoptotic cells in bioreactor cultures, which indicated a ‘good functioning’ of the cultured tissue. On the contrary, the lack of apoptotic cells in control cultures would be due to necrosis occurring between D50 and D90. Specific kinetic studies should be carried out to determine the optimal culture time for the best follicle yield. Besides, experiments in chitosan bioreactors with the addition of various growth factors should improve that yield.

One difficulty encountered in the present work was the distinct response to the culture of two ovaries out of five. The reason may lie in the pre-culture conditions (biopsy, conservation, transport, gestational age, etc.). Current investigations and a sensitivity analysis should clarify this issue. Furthermore, a number of interesting tests for follicle growth that were not carried out in the present study may be the diameters of oocytes (at the beginning and end of culture) as well as hormone assessments (estradiol, anti-mullerian hormone, etc.). The times at which these checks should be made are now under consideration.

## 5. Conclusions

Over three months, micro-bioreactor folliculogenesis outperformed traditional culture. Over time, growing intact primary and secondary follicles tended to increase in the bioreactor but decrease in control cultures. This is a promising result in the search for follicle preservation, simpler (one-step) or more efficient follicle culture, and studies on long-term exposure of follicles to various growth factors or drugs.

## Figures and Tables

**Figure 1 medicina-58-01565-f001:**
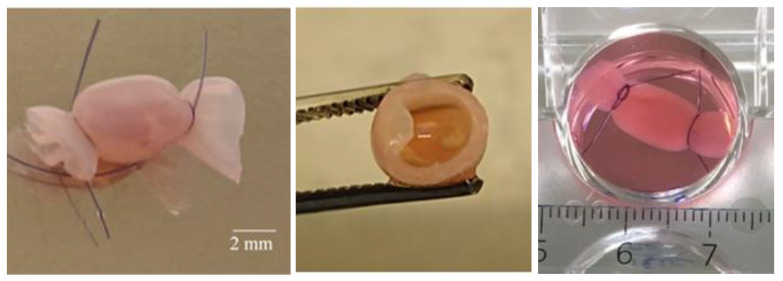
Chitosan bioreactor. From left to right: bioreactor containing ovarian tissue and tied at each end with suture thread; cross-section of the bioreactor showing the inner cavity and wall; culture well with bioreactor in culture medium.

**Figure 2 medicina-58-01565-f002:**
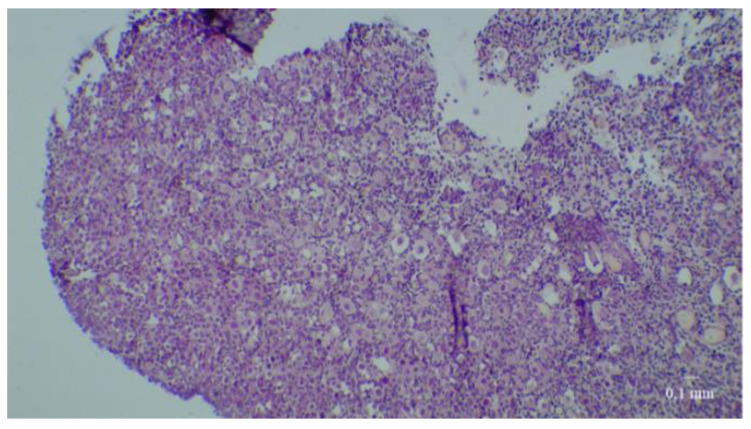
Fresh fetal ovarian tissue before culture. Slices colored before culture with hematoxylin-eosin-safran at low and high magnification.

**Figure 3 medicina-58-01565-f003:**
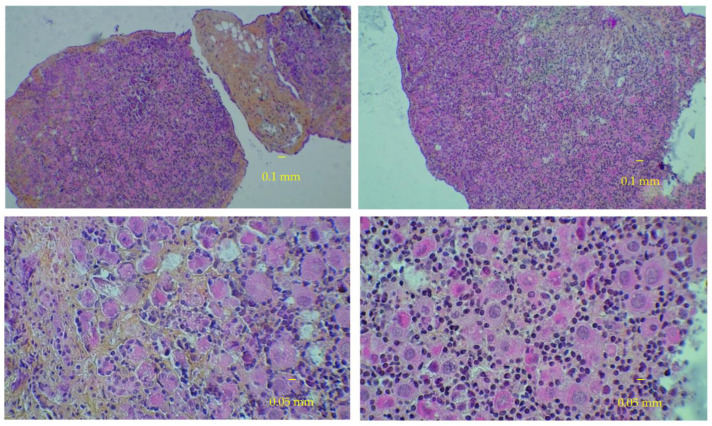
Fetal ovarian tissue in culture at day 50. Slices colored with hematoxylin-eosin-safran at low and high magnification. Left panels: control cultures. Right panels: bioreactor cultures.

**Figure 4 medicina-58-01565-f004:**
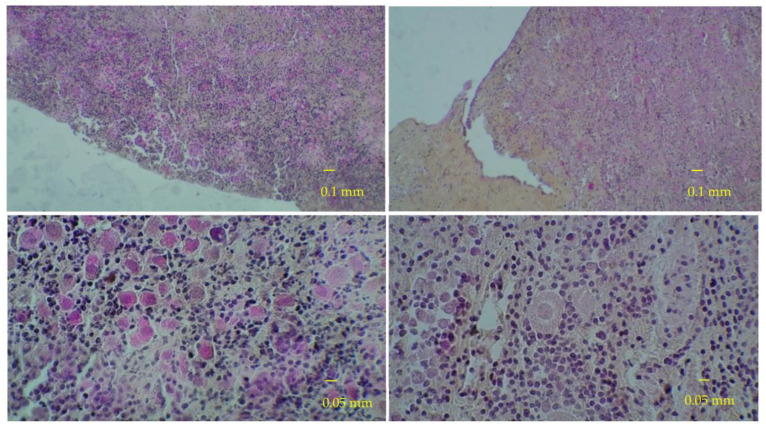
Fetal ovarian tissue in culture at day 90. Slices colored with hematoxylin-eosin-safran at low and high magnification. Left panels: control cultures. Right panels: bioreactor cultures.

**Figure 5 medicina-58-01565-f005:**
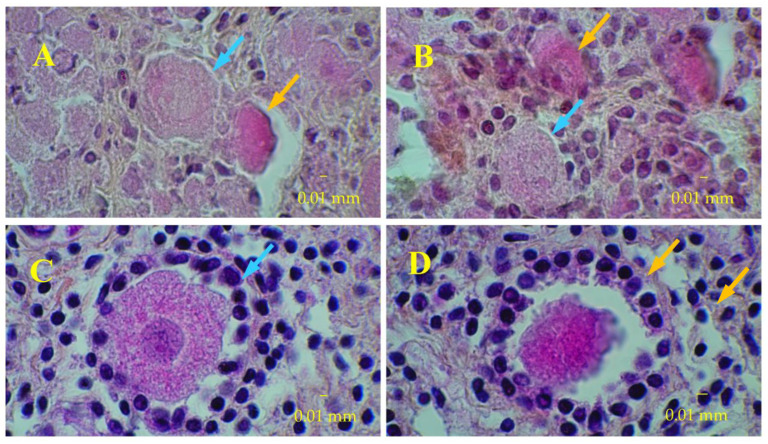
Fetal ovarian tissue in bioreactor at day 90. Slices colored with hematoxylin-eosin-safran. (**A**): primordial follicles. (**B**): primary follicles. (**C**): secondary follicle. (**D**): degenerated secondary follicle. Blue arrows indicate intact follicles. Orange arrows indicate degenerated follicles.

**Figure 6 medicina-58-01565-f006:**
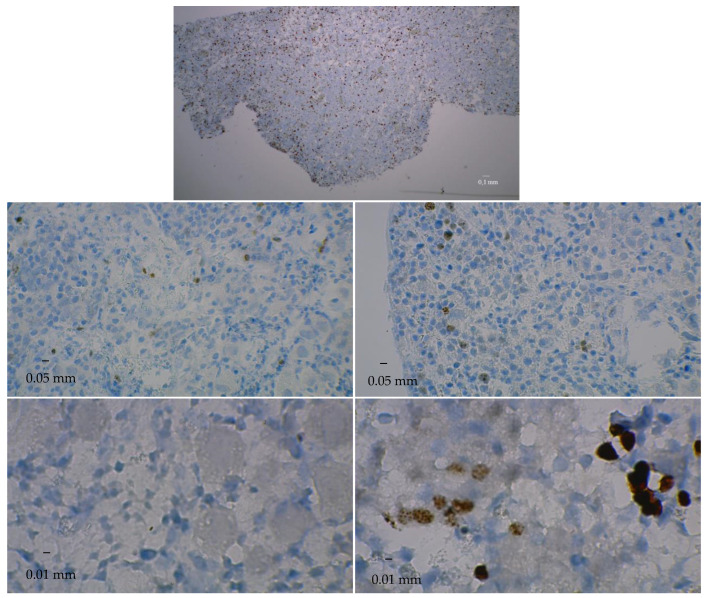
Ki67 immunostaining. Upper panel: Slice of fresh fetal ovarian tissue before culture. Middle row: Slices at day 50 of culture. Lower row: Slices at day 90 of culture. Left panels: control cultures. Right panels: bioreactor cultures. Positive (proliferating) cells have a brown color.

**Figure 7 medicina-58-01565-f007:**
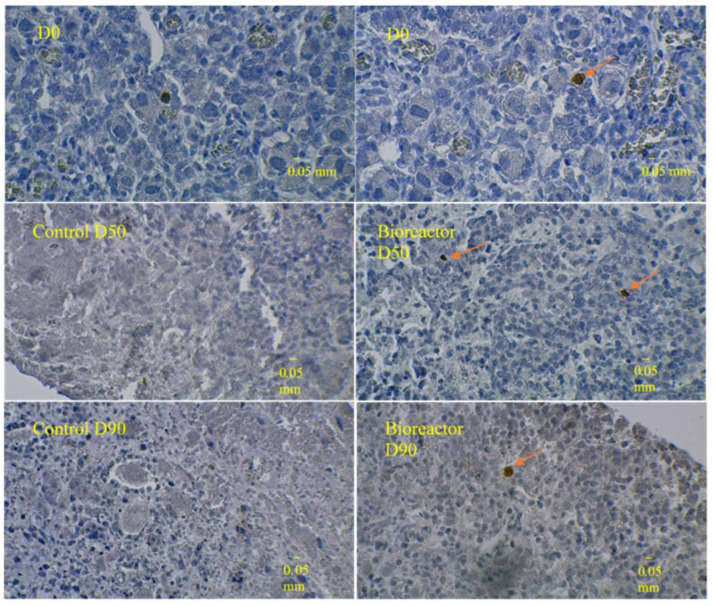
Cleaved caspase-3 immunostaining. Upper panel: Slice of fresh fetal ovarian tissue before culture. Middle row: Slices at day 50 of culture. Lower row: Slices at day 90 of culture. Left panels: control cultures. Right panels: bioreactor cultures. Apoptic cells—indicated by orange arrows—have brown cytoplasm.

**Table 1 medicina-58-01565-t001:** Percentages of growing follicles (pGF) relative to all intact follicles according to the culture time and type.

Ovary	pGF at D0	pGF at D50 in Bioreactor	pGF at D50 in Control Culture	pGF at D90 in Bioreactor	pGF at D90 in Control Culture
1	31.7	60	16.7	51	82.4
2	9.7	24.3	11.4	39.7	48.5
3	2.5	1.2	1.1	1.8	1.3
4	0	12.5	–	35.3	0
5	0	0	–	25	–

**Table 2 medicina-58-01565-t002:** Percentages of growing follicles (pGF) relative to all follicles present according to the culture time and type.

Ovary	pGF at D0	pGF at D50 in Bioreactor	pGF at D50 in Control Culture	pGF at D90 in Bioreactor	pGF at D90 in Control Culture
1	29.3	35.7	6.1	30.9	18.2
2	8.2	18.4	7.8	27.8	20.9
3	1.3	0.2	0.4	1	0.4
4	0	4.4	0	27.3	0
5	0	0	0	8.1	0

**Table 3 medicina-58-01565-t003:** Densities of proliferating and apoptotic cells according to the culture type and time.

	Day 50	Day 90
*Proliferating cells*		
Bioreactor cultures	14.1 ± 7.5 ^a^	10.5 ± 6.4
Control cultures	3.7 ± 3.6	0.0 ± 0.0
*Apoptotic cells*		
Bioreactor cultures	6.8 ± 2.2 ^b^	2.2 ± 1.1
Control cultures	0.0 ± 0.0	0.0 ± 0.0

Each value is the mean ± standard deviation of the number of ^a^ Ki67-positive cells/mm^2^ and ^b^ cleaved caspase-3-positive cells/mm^2^.

## Data Availability

The data are available from the first author upon written motivated request and commitment to keep the obtained data confidential.

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
