# Peer review of "Development and Survival of Human Ovarian Cells in Chitosan Hydrogel Micro-Bioreactor"

_medicina, 2022, doi:10.3390/medicina58111565_

Round 1
Reviewer 1 Report (Previous Reviewer 2)
Given the revision performed by the authors, the manuscript canbe considered for publication in its present form
Author Response
The authors do thank sincerely the reviewer for his kind collaboration and, especially, for easing the publication of this interesting work.

Reviewer 2 Report (New Reviewer)
In this paper, the authors evaluated if primordial follicles cultured within human ovarian cortex are able to develop to secondary stage when pieces of ovarian cortex were inserted into a chitosan hydrogel microbioreactor and cultured in vitro. After 50 or 90 days, a higher percentage of growing follicles was detected in bioreactor in comparison with control. Although the study is of interest, I have some questions that must be addressed before accepting the paper.
1. Experimental data are not clearly expressed. The different percentages of growing follicles between bioreactor and control is not sufficient to sustain conclusions. Data as mean follicle and oocyte diamenters at the beginning and at the end of culture are necessary; moreover, evaluation of estradiol and AMH release during culture is generally used as both are biomarkers of preantral follicle growth. Also data on PI3K/PTEN/AKT/FOXO3a pathway should be useful to confirm the effective growth of follicles.
2. It is unclear the reason why data from tables 1 and 2 have been plotted also in the figure 5 A-D.
3. l 335-339. Activation rate of primordial follicles is strongly influenced by culture conditions (see Shah JS et al JARG 2018, 35:1135). The authors should reduce hetogenicity of their samples by selecting for cultures pieces of ovarian cortex with similar morphological characteristics, because fragmentation per se can promote activation of primordial follicles. I suggest the authors to increase the number of cultured fragments to increase significativity of their results.
Author Response
In this paper, the authors evaluated if primordial follicles cultured within human ovarian cortex are able to develop to secondary stage when pieces of ovarian cortex were inserted into a chitosan hydrogel microbioreactor and cultured in vitro. After 50 or 90 days, a higher percentage of growing follicles was detected in bioreactor in comparison with control. Although the study is of interest, I have some questions that must be addressed before accepting the paper.
The authors thank the reviewer for his/her attentive examination of the manuscript.
1- Experimental data are not clearly expressed. The different percentages of growing follicles between bioreactor and control is not sufficient to sustain conclusions. Data as mean follicle and oocyte diameters at the beginning and at the end of culture are necessary; moreover, evaluation of estradiol and AMH release during culture is generally used as both are biomarkers of preantral follicle growth. Also, data on PI3K/PTEN/AKT/FOXO3a pathway should be useful to confirm the effective growth of follicles.
The authors do appreciate the above-mentioned suggestions for confirmation of the study findings. First, they would like to underline that, in this pilot study, multiplying the number of checks of various natures was not practicable because of limited financial and human resources, and, especially, because of the bulk and length of the experimental process. Indeed, one essential culture parameter the investigators wanted to explore first was the possible length of culture duration. Now that the resistance to culture is more or less known, it becomes possible to carry out confirmation tests for growth and/or stages.
To reflect this, the following was added to the Discussion:
“A number of interesting tests for follicle growth that were not carried out in the present study may be the diameters of oocytes (at beginning and end of culture) as well as hormone assessments (estradiol, anti-mullerian hormone, etc.). The times at which these checks should be made are now under consideration.”
2- It is unclear the reason why data from Tables 1 and 2 have been plotted also in Figure 5 A-D.
We initially thought that Figure 5 would display a clearer visual idea (vs. tables) about the states of the cultures. Now, we do not mind removing it from the paper (which is done).
3- Lines 335-339. Activation rate of primordial follicles is strongly influenced by culture conditions (see Shah JS et al., JARG 2018; 35:1135). The authors should reduce heterogeneity of their samples by selecting for cultures pieces of ovarian cortex with similar morphological characteristics, because fragmentation, per se, can promote activation of primordial follicles. I suggest the authors to increase the number of cultured fragments to increase the significance of their results.
As partially mentioned above, multiplying the number of samples was not practicable because of i) limited sample sources; ii) limited financial and human resources; and, especially, iii) the bulk and length of the experimental process. Currently, ‘shortcuts’ are being examined and will probably allow increasing the number of initial samples and reduce sample heterogeneity.
The authors do not deem the promotion of activation via simple tissue fragmentation a problem or a limitation. The main issue was not the cause of activation initiation but the continuation and duration of the cultures in the new system as well as the oocytes’ outcomes. This is now mentioned in the Discussion.
“The initiation of oocyte activation could have been tissue fragmentation by itself (Shah et al.). However, this might not be a serious problem because the aim of the experiments was the length and issue of cultured oocytes rather than the cause or promotion of their activation.”

Reviewer 3 Report (New Reviewer)
In this manuscript, the authors attempted to test the long-term ability of human ovarian cortex cells to develop in unconventional culture conditions. The topic is exciting and may be clinically applicable soon. However, several points need clarification.
Abstract:
Line 22-23: A total of five ovarian cortex tissue were used for this study. The “(35.6±10.4) week” may not be suitable for describing such a small sample size distribution. The range between the minimal and maximal weeks may be better used here. In addition, the term “embryos” is used for smaller than 14 days (2 weeks) of development. The term “fetuses” is better than “embryos” in this manuscript.
Line 29: It’s difficult to read the sentence “of bioreactor cultures (16.12% difference at D50 but only 0.12 difference at D90”. Please keep the unit or scale of numbers consistent within the same manuscript.
Results:
Line 270-271: Although the authors stated that “Fetal ovarian tissue histology slices that illustrate these trends are given in figures 6 and 7,” the histology in figures 3 and 4 is more suitable for the trend in figure 5. By contrast, the histology findings in figures 6 and 7 are summarized in Table 3. Please clarify it. Furthermore, could you provide the histology with Caspase-3 staining?
Discussion:
Line 330 and 335: The term “the latter” makes the sentence difficult to understand. Please clarify them.
One of the exciting findings in this manuscript is that ovarian cells in the bioreactor demonstrate more apoptotic cells compared to those in the control culture. Please explain and discuss this finding and the possible mechanism and effect on oocyte or follicular development.
Author Response
In this manuscript, the authors attempted to test the long-term ability of human ovarian cortex cells to develop in unconventional culture conditions. The topic is exciting and may be clinically applicable soon. However, several points need clarification.
Abstract:
Line 22-23: A total of five ovarian cortex tissue were used for this study. The “(35.6±10.4) week” may not be suitable for describing such a small sample size distribution. The range between the minimal and maximal weeks may be better used here. In addition, the term “embryos” is used for smaller than 14 days (2 weeks) of development. The term “fetuses” is better than “embryos” in this manuscript.
In this pilot type of culture of fetus-originated cells, multiplying the number of samples was not practicable because of i) limited sample sources; ii) limited financial and human resources; and, especially, iii) the bulk and length of the experimental process. Currently, ‘shortcuts’ are being examined and will probably allow increasing the number of initial samples.
Regarding the presentation of fetuses ages, the mean ± SD is now replaced by the age range (23 to 39 weeks gestation). Besides, all occurrences of “embryo” were examined for a necessary replacement by “fetus”.
Line 29: It’s difficult to read the sentence “of bioreactor cultures (16.12% difference at D50 but only 0.12 difference at D90”. Please keep the unit or scale of numbers consistent within the same manuscript.
Thank you for pointing out this inconsistency. The mention of this difference is now amended as follows: “… in favor of bioreactor cultures (16.12% difference at D50 but only 0.12% difference at D90)” as already mentioned in the last paragraph of section 3.1.
Results:
Line 270-271: Although the authors stated that “Fetal ovarian tissue histology slices that illustrate these trends are given in figures 6 and 7,” the histology in figures 3 and 4 is more suitable for the trend in figure 5. By contrast, the histology findings in figures 6 and 7 are summarized in Table 3. Please clarify it. Furthermore, could you provide the histology with Caspase-3 staining?
We do agree with the above statements. To consider this, we have decided to delete Figure 5, and cite Table 3 together with Figures 5 and 6 (previous Figures 6 and 7).
The micrographs relative to caspase-3 staining are now added as a new Figure 7.
Discussion:
Line 330 and 335: The term “the latter” makes the sentence difficult to understand. Please clarify.
This sentence is now amended as follows: “One explanation for that higher density of apoptic cells is that…”.
One of the exciting findings in this manuscript is that ovarian cells in the bioreactor demonstrate more apoptotic cells compared to those in the control culture. Please explain and discuss this finding and the possible mechanism and effect on oocyte or follicular development.
Potential explanations for the fact that there were more apoptotic cells in the bioreactor vs. the control cultures are exposed in the third and sixth paragraph of the Discussion (One explanation for that higher density of apoptic cells is that…” and “One interesting fact was the presence of apoptotic cells…”
We have explored the recent literature on similar cultures and found no similar result; thus, no other possible explanations by other authors.

Round 2
Reviewer 2 Report (New Reviewer)
The authors have satisfactorily answered my questions
Reviewer 3 Report (New Reviewer)
The authors have adequately responded to my comments. Now, this manuscript is acceptable for publication.
This manuscript is a resubmission of an earlier submission. The following is a list of the peer review reports and author responses from that submission.
Round 1
Reviewer 1 Report
I read the article entitled ‘Development and survival of human ovarian cells in chitosan bioreactor’.
Introduction
Please don’t use ‘team’. Instead of please add ‘et al.’
What is large animal?
The sentence ‘Fresh ovaries (n=5) from human embryos aged 35.6 ± 10.4 weeks amenorrhea’ is incomprehensible.
Ethical consent form and number of the approval should be stated.
Please add the references for hypothesis (One hypothesis for the latter fact is that, in control cultures, follicles are altered but not eliminated, whereas, in bioreactor cultures, altered follicles are eliminated and, there- fore, are no longer visible on histological examination. Another hypothesis may be the 326 wide heterogeneity of the ovarian tissues cultured; those put in the bioreactors would be less rich in follicles than those put in traditional medium….)
As a result of this study, is there a result that mature oocytes are not obtained?
It is recommended that the article be evaluated by both histology-embryology specialist and biomedical engineering specialist.
Yours sincerely,
Author Response
Introduction
- Please don’t use ‘team’. Instead of please add ‘et al.’
All uses of “team” are now replaced by “et al.” as suggested.
- What is “large animal”?
We apologize for this error. This has been replaced by “farm animals”.
Materials and methods
- The sentence ‘Fresh ovaries (n=5) from human embryos aged 35.6 ± 10.4 weeks amenorrhea’ is incomprehensible.
This reads now: “Fresh ovaries (n=5) extracted from 33.6(± 6.6)-week-old human fetuses”. As usual the values are the mean and the standard deviation.
- Ethical consent form and number of the approval should be stated.
All ethical aspects are detailed in the last section of M&M.
Discussion
- Please add the references for hypothesis (“One hypothesis for the latter fact is that, in control cultures, follicles are altered but not eliminated, whereas, in bioreactor cultures, altered follicles are eliminated and, therefore, are no longer visible on histological examination. Another hypothesis may be the wide heterogeneity of the ovarian tissues cultured; those put in the bioreactors would be less rich in follicles than those put in traditional medium…”)
The above-cited hypotheses are not extracted from the literature but are the manuscript authors’ own attempts for explanation. All occurrences of “hypothesis” have been examined and replaced by “explanation” when needed.
- As a result of this study, is there a result that mature oocytes are not obtained?
In our work, we reached the stage of secondary follicle. Obtaining mature oocytes requires further in vitro culture steps.
- It is recommended that the article be evaluated by both histology-embryology specialist and biomedical engineering specialist.
The article has been already read and intensively corrected by a specialist in biotechnologies and polymer science. Nevertheless, we welcome another evaluation and are willing to answer other specialists’ questions.
Yours sincerely,

Reviewer 2 Report
I read with great interest the current study on the long term culture of human ovarian cortex in a chitosan bioreactors compared to conventional conditions. The authors observed a higher percentage of growing primary and secondary follicles at D50 and D90 in bioreactor cultures than in control cultures. This strategy is intriguing to test new opportunity in fertility preservation. The manuscript is clear, even requiring english grammar revision.
I would suggest to clarify whether the authors decided to set follicular growth evaluation in the two time points D50 and D90.
Did the authors evaluate vascularization too?
Author Response
I read with great interest the current study on the long-term culture of human ovarian cortex in a chitosan bioreactors compared to conventional conditions. The authors observed a higher percentage of growing primary and secondary follicles at D50 and D90 in bioreactor cultures than in control cultures. This strategy is intriguing to test new opportunity in fertility preservation. The manuscript is clear, even requiring English grammar revision.
The whole manuscript has been revised for English language improvement.
I would suggest to clarify whether the authors decided to set follicular growth evaluation in the two time points D50 and D90.
Thank you for pointing out this issue. Indeed, the two time points were decided since the protocol design: 90 days is the time needed for full in vivo folliculogenesis (Gougeon) and 50 days is nearly the mid-time for that process.
Did the authors evaluate vascularization too?
No. Examining vascularization was not originally planned. However, we believe this examination will be essential once the process for obtaining mature follicles will be reliably reproducible.
